# How do cohabitation and marital status affect mortality risk? Results from a cohort study in Thailand

Jiaying Zhao,[1] Chi Kin Law,[2] Matthew Kelly ,[3] Vasoontara Yiengprugsawan,[4] Sam-Ang Seubsman,[5] Adrian Sleigh [6]

**Correspondence to**
Matthew Kelly;
Matthew.Kelly@anu.edu.au

## ABSTRACT

**Objective** To examine the relationship between baseline union status (ie, including marriage and cohabitation) and mortality, paying attention to gender differentials, through an 11-year follow-up of a large cohort in Thailand.

**Design** Cohort data from Thai Cohort Study (TCS) were linked official death records over an 11-year follow-up period.

**Setting** Community-based adults in Thailand.

**Participants** 87 151 Thai adults participated in TCS cohort.

**Method** Cox regression models measured longitudinal associations between union status and 11-year mortality.

**Results** From 2005 (baseline) to 2016, persons who cohabited and lived with a partner, married persons but not living with a partner and separated/divorced/widowed people were more likely to die compared with those married and living together with a partner. Those who did not have good family support had a higher death risk than those having good family support.

Single or cohabiting women had higher risks of mortality than women who were married and living together with a partner throughout follow-up, while separated/divorced/widowed men had higher risks of mortality than counterpart males.

**Conclusions** Our study reveals the protective effect of marriage and living together on mortality in Thailand, an understudied setting where institutionalisation of cohabitation is low leading to a limited mortality protection. Public policies for moderating mortality should thus be gender nuanced, culturally and institutionally specific. Also, we demonstrate that in settings such as Thailand, where marital status is not always defined in the same way as in western cultures, the need to measure cohabitation in locally relevant terms is important.

## STRENGTHS AND LIMITATIONS OF THIS STUDY

⇒ This is a large nationwide cohort (ie, 87 151 participants) with 11-year follow-up period produced reliable estimate of effect sizes while also measuring the longitudinal effects of changes in union status on mortality.

⇒ The large variety of variables collected in the study allow for control of multiple potential confounders.

⇒ Cause of death was not able to be included in the analysis limiting the ability to interpret the findings.

⇒ · The difficulty in defining marriage and cohabitation in Thailand may indicate there was some misclassification of individuals making the cohabitation—mortality relationship difficult to measure.

Two major processes have been proposed to explain the marital status differences in mortality risks: (1) selection mechanisms and (2) protection mechanisms.[1 8] Selection theory suggests that unhealthy people ('confounding variable') may be selected outside the bounds of marriage into other union types, such as cohabitation.[9] The 'protection mechanisms' theory suggests that marital status has a causal impact on mortality risk reduction through the provision of social/economic support, through improvement of unhealthy behaviours and mental health ('intermediate variable'), and provision of health information.[1 3 6 10 11] A meta-analysis indicates that support from family was more beneficial than support provided by friends in reducing mortality.[6] A further clarification of the interaction between family support and union status and their effects on mortality can improve our understanding of the pathways between union status and mortality risk reduction.

Cohabitation may facilitate similar functions as marriage, though such a protection effect may be weaker than marriage and may vary across populations.[9 12 13] Some evidence from developed countries shows that cohabiting persons have lower mortality than those

## INTRODUCTION

Inequalities of mortality risk across categories of marital status have been well documented by previous longitudinal studies.[1–3] Married persons generally have lower mortality than those who are never married, divorced, separated or widowed in both developed[4] and developing countries.[5] Such patterns were found to be more marked among men than women.[6 7]

living alone, particularly for men and a combination of married and cohabited state may better predict mortality than marriage alone.[13 14] Similar to married status, cohabitation may improve health through emotional and instrumental support from partners.[15] But cohabitation, with lower levels of commitment, support and stability than marriage, could constrain economic, social and psychological resources.[9 15 16] Also, living arrangements, incorporating the partnership situation and household composition, was associated with mortality.[17 18] Evidence on the health effect of cohabitation, particularly incorporating living arrangements is inconsistent and sparse in low-income and middle-income countries.

Gender differences could be another potential moderator for the effect of union status on mortality.[3 14] Men's mortality may profit more from union status than women.[3 13 19 20] Men may benefit more from risk behaviour reduction and emotional support than women,[19 21] while women may benefit more financially.[1 22] After union disillusion, excess mortality is greater among men than women.[20 23] This may be because women are more likely to adopt the role of providing emotional support and performing household tasks for their spouse. Men may suffer more once union status dissolves[13] but separated/divorced/widowed women may feel relieved of their emotional duty to care, and consequently lower their mortality.[24]

Gender differences in the effect of union status on mortality may vary in societies with different gender role norms (eg, the male breadwinner model).[24] In societies with a low degree of institutionalisation of cohabitation, the mortality benefit from union status due to financial support may be weaker for cohabited women than married women. Cohabited women may also benefit less emotionally from union status due to low social acceptance of cohabitation.

Cohabitation has become more common in Asian countries.[25 26] In Thailand, the proportion of cohabiting individuals increased from 6% in 2005–2009 to 9% in 2010–2014.[27] However, cohabitation may have different social, cultural and institutional interpretations as well as legal rights for relationships in Eastern societies.

Thai society has strong religious identification,[28] and more traditional cultural norms around cohabitation.[29] The strong religious identification has a role in the production of gender norms and expectations.[30] However, during the past decades, attitudes towards marriage have been changing.[31 32] Increased female workforce participation, work related pressures, high costs of living and an increasing burden of care giving for elderly parents have led to delayed marriages and downward pressure on marriage rates.[31 33]

Reported cohabitation rates remained low in Thailand, however, this may relate to the inexact definition of marriage in Thai society. State registration of marriage is practised in Thailand but those married through traditional Buddhist religious ceremonies are also considered married. In 2005, around 34% of currently married people were not registered, and the proportion was higher for those aged 15–34 years.[34] Overall, cohabiting without marriage in Thailand is likely to be more common than surveys reveal.[31] Younger, more educated and Bangkok dwelling Thais are more likely to cohabit with their partner.[25]

In many Western countries de facto partnering confers rights and obligations on partners that are identical to those married. Those rights include economic security and sharing of marital property in the event of divorce.[35] However, cohabitation in Thailand does not incur the same legal outcomes as registered marriages in terms of sharing property and other assets.[36]

Variation in different social, cultural and institutional interpretations as well as legal rights for relationships may lead to different associations between cohabitation and mortality.[6 37 38] The gap in mental health between married and cohabiting individuals is larger in countries with a low degree of institutionalisation of cohabitation. In such countries the following features are expected: (1) cohabitation is relatively uncommon and low in social acceptance[38]; (2) traditional gender role persists[37] and (3) and religiosity is high.[37] Disapproval of cohabitations and uncertainty of future may lead to lower level of social support and poorer mental well-being for cohabiters in such society.[38]

Previous studies examined the influence of marital status on mortality in Eastern societies but did not differentiate between cohabitation and marriage.[39] It remains unclear whether cohabitation affords Eastern populations' protective benefits similar to marriage.

This study aims to fill this knowledge gap by examining the relationship between baseline union status (ie, including marriage and cohabitation) and mortality, paying attention to gender differentials, through an 11-year follow-up of a large cohort in Thailand. The potential mechanism linking union status to mortality from socioeconomic, psychological, and behavioural aspects will be investigated.

### Hypotheses

Considering the institutional and cultural features around marriage in Thailand, we test the following hypotheses:
1. People who were single, or separated/divorced/widowed have higher mortality than married people living together.
2. Cohabiting people living with a partner have a higher mortality risk than married people living together, but may have lower mortality than single, or separated/divorced/widowed people.
3. People who have stronger family support have lower mortality, and this effect cannot be fully explained by differentials in union status.
4. 'Separated/divorced/widowed' males have a higher mortality risk compared with married men living together with a partner, while such higher mortality risk may be moderated among females.

5. Cohabiting women living with a partner have a higher mortality risk compared with married women living together with a partner, while such higher mortality risk may be limited among males.

## METHODS

### Study population and study design

Data from a large nationwide longitudinal study of health dynamics in Thailand—the Thai Cohort Study (TCS)—were linked with official death records over an 11 year follow-up period. The baseline cohort was recruited in 2005 with a questionnaire mailed to all current students of Sukhothai Thammathirat Open University (STOU); 87 151 responded, covering 43.5% of the STOU students at that time. STOU students are off-campus adult distance-learners residing throughout the country. Characteristics of TCS members are similar to the overall Thai population, in terms of social geography, religion, socioeconomic status and income, but members are in general more highly educated and more urban.[40] In 2005, at the study baseline, the cohort and the general Thai population shared a median age of 29 years, median income of around US$2500, were 94% Buddhist, and just over 50% female, and had a similar regional distribution across the country.[40 41]

### Measurements

#### Mortality

Mortality status was traced with death registration records from the Thai Ministry of Interior covering the period from TCS baseline in 2005 to November 2016. The current death registration system covers about 95% of all deaths in Thailand.[42] Data linkage was conducted by matching the unique citizen identification number, which cohort members provided to the study at baseline. By November 2016, 1401 deaths were recorded within the cohort.

#### Union status

Using the TCS baseline data, union status was categorised by the following three questions: (1) 'What is your current status?' (single, living with partner or married). Those who responded 'married' were further classified by question: (2) 'What is your current marital status?' Those who responded 'married—first and only marriage' or 'remarried—second or later marriage' were classified as 'married', while those who responded 'separated from someone you have been married to (but not divorced)', 'divorced' or 'widowed' were classified as 'separated/divorced/widowed'. We further classified union status with living arrangement by 3. 'Do any of the following people (ie, spouse/partner) usually live in your home?'

Conjugal union status was coded into seven categories: (1) single, (2) cohabited and living with a partner, (3) cohabited but not living with a partner, (4) married but not living together with a partner, (5) married and living together with a partner (the reference group),

(6) separated/divorced/widowed and (7) separated/divorced/widowed but living with a partner.

#### Family support

Family support was assessed by the following question using a four-point scale: 'How would you rate the support you are getting from your family?' The proportion of deaths during 11-year follow-up for those reporting 'quite a bit of support', and 'a lot of support' was 1.65% and 1.45%, while that for those reporting "very little', 'little' or 'not relevant' was 2.67%, 2.24% and 3.75%, respectively. The statistical power may be influenced due to the small number of deceased participants for those with little (128), very little (57) support and not relevant (23), and a merger of the three groups enables a more robust statistical analysis in this study. Therefore, those who responded 'very little', 'little' or 'not relevant' were coded as not having good family support, while those who responded 'quite a bit of support' and 'a lot of support' were coded as having good family support.

#### Other baseline covariates

A wide range of covariates were available including baseline variables describing sex, birth year, urban or rural residence, and income, smoking status and alcohol consumption at baseline, baseline pre-existing psychological and physical health conditions and baseline body mass index (BMI). Details of each covariate are shown in table 1.

We also tested social support in the model. Social support was measured using the question 'How would you rate the support you are getting from the following: government, religious groups, and good friends/neighbours/people in workplace'. Those who responded 'very little' or 'little', 'not relevant' were classified as not having good social support for that question. But these covariates were not significantly associated with mortality; and we did not report it in the final result.

#### Statistical analyses

Our longitudinal analyses of all-cause mortality continued until the end of the observation period in November 2016. The exposure variables of interest were union status and family support. We tabulated November 2016 survival status by union status, family support and all covariates.

We examined links between mortality as the outcome and union status and family support as the mutually adjusted exposure variables of interest. We used Cox regression to estimate multivariate HRs for men and women and their 95% CIs given the assumption of proportionality of hazards were held. Covariates were included in sequence as models were developed incrementally.

To examine the effect of other covariates on the association between baseline union status and the subsequent mortality outcome, four regression models were compiled. The included covariates and the exposure variables of interest for each model are shown in table 2. The variance inflation factors for all covariates and the

**Table 1** The measurements and classification of baseline (2005) covariates

| Variable | Measurement question | Original response category | Analytical category |
|---|---|---|---|
| Sex | You are: | Male; female | Male; female |
| Birth year | In what year were you born? | Year of birth | 1959; 1960–1969; 1970–1974; 1975–1979; 1980 |
| Residence | Where is your current permanent home located now? | Countryside; city/town | Urban; rural |
| Income | What is your average personal monthly income? | Less than 3000 Baht; 3001–7000 Baht; 7001–10 000 Baht; 10 001–20 000 Baht ; 20 001–30 000 Baht; over 30 000 Baht | Personal monthly income: ≤Baht7000, Baht7001–Baht20 000 or >Baht20 000 (US$1~Baht30) |
| Smoking status | 1. Have you ever smoked? 2. If you have quit smoking, at what age did you stop? | 1. Yes; No (will categorised 'Never'). 2. XX years; still smoking (will be categorised as 'current smoker'), otherwise will be categorised as 'former smoker', | Never; former smoker; current smoker |
| Alcohol consumption | Have you ever drunk alcohol? | Occasional social drinker; No, never; Current regular drinker; Used to drink before, now stopped | Never drinker; occasional social drinker; current regular drinker; former drinker |
| Pre-existing physical conditions | The next questions ask about some health conditions whether you ever been told by a doctor that you have this condition. (please tick all that apply) | Diabetes (needing insulin); Diabetes (do not need insulin); liver disease (not cancer); Kidney disease; High cholesterol/high blood lipids; High blood pressure; ischaemic (coronary) heart disease; cerebrovascular disease (stroke); liver cancer; lung cancer; cancer of the digestive system; Breast cancer; other cancers | Whether a doctor had ever diagnosed: diabetes, cancer, cardiovascular diseases, kidney and liver diseases |
| Pre-existing psychological conditions | The next questions ask about some health conditions whether you ever been told by a doctor that you have this condition. (please tick all that apply) | Depression/anxiety | Whether a doctor had ever diagnosed psychological conditions (ie, depression or anxiety) |
| Body mass index (BMI) | 1. What is your weight? 2. What is your height without shoes? | 1. XXX kg 2. XXX cm | Calculated by validated self-reported height and weight (Lim et al[60]). Asian BMI standards: underweight (<18.5), normal (18.5 to <23), overweight at risk (23 to <25), obese I (25 to <30), and obese II (≥30) (Kanazawa et al[61]) |

BMI, body mass index.

exposure variables were less than 4, indicating no multicollinearity issues in the models.[43] We further applied the above four models by sex.

From model 1 to model 3, we gradually added confounding variables at individual, family and socioeconomic levels. In model 4, we included health behaviour and pre-existing psychological and physical conditions and BMI. These health behaviour and status variables may serve as 'intermediating factors' for the effect of union status on mortality if the HR for union status changed significantly when these variables were added in the Model. If HR for union status changed little, these health behaviour and status variables may act as confounding variables for the effect of union status on mortality.[44]

**Patient and public involvement statement**

Study participants were all enrolled students at an Open University in Thailand. Research questions and approaches were first piloted with these students and the study design was adjusted accordingly. At various stages through the research project participants were provided with summary results of the study. As well the lead investigator made frequent public appearances

| Table 2 | Variables included in the analysed models |
|---------|-------------------------------------------|
| **Model** | **Variables included** |
| 1 | Baseline union status and demographic variables (ie, sex, year of birth, residence). |
| 2 | Variables in model 1, and family support. |
| 3 | variables in model 2, and socioeconomic variables (ie, incomeV). |
| 4 | Variables in model 3, and related health behaviours (ie, smoking status, alcohol consumption), pre-existing psychological and physical conditions and BMI. |

BMI, body mass index.

on university radio and television media to discuss the results of the study.

## RESULTS

At 2005 baseline, 49.3% of cohort members were single, 4.9% were cohabited and living with a partner, 1.4% were cohabited but not living with a partner, 3% were married but not living with a partner, 35.3% were married living together with a partner and 3.7% were separated/ divorced/widowed and not living with a partner, 0.6% were separated/divorced/widowed but living with a partner. There were 89.7% members reporting have good family support. The relationship between baseline union status, family support and a range of potential confounders were shown in table 3.

### Union status, family support and mortality

From 2005 (baseline) to 2016, separated/divorced/ widowed people and married not living with a partner were more likely to die than married people living with a partner, but singles and those who cohabited were less likely to die compared with married people living together with a partner (table 4). Those not having good family support were more likely to die than those having family support. Males, older birth cohorts, those living in rural areas, current smokers, current regular drinkers, those having psychological and chronic physical disease, and obese people were more likely to die.

In the Cox proportional hazard model (table 5), after controlling for sex, birth cohort, rural or urban residence, singles, persons who cohabited and living with a partner, married persons but not living with a partner, and separated /divorced/widowed people were more likely to die compared with those married living together with a partner (model 1). Further adjustment of family support (model 2) reduced the corresponding HR. Those who do not have good family support had a higher death risk than those having good family support. Overall, no significant interaction was found between union status and family support (p>0.05) (online supplemental appendix tables 1 and 2). We stratified by whether participants had good family support at baseline. Among respondents who

had poor family support, no mortality risk difference was found in those cohabited and living with a partner compared with those who married living together with a partner.

When the effect of personal income was included (model 3), the HR reduced but remained significant for singles, cohabited and living with a partner, and separated/divorced/widowed people, but the HR for those married but not living with a partner became not significant. HR for family support did not change significantly when income was included. When health behaviours, pre-existing psychological conditions and chronic conditions (model 4) were included, the HR for union status and family support changed little.

In the final model 4, certain covariates also associated with mortality risk were included. Males, older birth cohort members, rural residents, persons with low income, smokers, current regular drinkers or those who stopped drinking, and those having psychological or physical conditions were all more likely to die, compared with their counterparts (online supplemental appendix table 1)

### Sex differentials in the effect of union status on mortality

We repeated the models by sex with results shown in table 5. Males and females had a different mortality differential pattern by union status. After controlling for birth cohort, rural or urban residence, both single men and women had higher risks of dying than their counterparts who have married living together with a partner. However, cohabited women living with a partner had higher risks of dying than married women living together with a partner, while such increase in mortality risk was not significant among men. By contrast, separated/divorced/widowed men had a higher mortality risk than counterpart men, while such risk did not significantly increase among separated/divorced/widowed women (model 1). Further adjustment for family support (model 2) reduced the corresponding HR. The HR for those not having good family support for females was higher than males.

When income was included in model 3, the HRs for single and separated/divorced/widowed men reduced, but remained similar for single women. The HRs for cohabited women living with a partner was similar to those in model 2, while those for men became not significant. HR for family support did not change significantly when income was included. In the final model 4, when health behaviours, pre-existing psychological and chronic conditions and BMI were included HR for union status and family support changed little in these models.

In the final model 4, single women, cohabited women living with a partner, and separated/divorce/widowed men had significantly higher mortality risks than their married counterparts living together with a partner.

### Effect of union status on mortality by age group

A further subgroup analysis by age group was conducted (see online supplemental appendix table 2). Overall,

**Table 3** Sociodemographic attributes at baseline (2005) by union status in the Thai Cohort Study

| | Single | (%) | Cohabited and living with a partner | (%) | Cohabited but not living with a partner | (%) | Married but not living with a partner | (%) | Married and living with a partner | (%) | Separated/ divorced/ widowed | (%) | Separated/ divorced/ widowed but living with a partner | (%) | Missing | (%) | Total | (%) |
|---|---|---|---|---|---|---|---|---|---|---|---|---|---|---|---|---|---|---|
| Total | 42931 | | 4235 | | 1201 | | 2624 | | 30721 | | 3206 | | 566 | | 1667 | | 87151 | |
| **Sex** | | | | | | | | | | | | | | | | | | |
| Males | 16649 | 38.8 | 1983 | 46.8 | 517 | 43.0 | 1340 | 51.1 | 16759 | 54.6 | 1086 | 33.9 | 345 | 61.0 | 806 | 48.4 | 39485 | 45.3 |
| Females | 26282 | 61.2 | 2252 | 53.2 | 684 | 57.0 | 1284 | 48.9 | 13962 | 45.4 | 2120 | 66.1 | 221 | 39.0 | 853 | 51.2 | 47658 | 54.7 |
| Missing | 0 | 0.0 | 0 | 0.0 | 0 | 0.0 | 0 | 0.0 | 0 | 0.0 | 0 | 0.0 | 0 | 0.0 | 8 | 0.5 | 8 | 0.0 |
| **Birth year** | | | | | | | | | | | | | | | | | | |
| 1959 | 584 | 1.4 | 48 | 1.1 | 10 | 0.8 | 304 | 11.6 | 3799 | 12.4 | 460 | 14.3 | 107 | 18.9 | 74 | 4.4 | 5386 | 6.2 |
| 1960–1969 | 3404 | 7.9 | 399 | 9.4 | 76 | 6.3 | 732 | 27.9 | 11596 | 37.7 | 1353 | 42.2 | 260 | 45.9 | 217 | 13.0 | 18037 | 20.7 |
| 1970–1974 | 5248 | 12.2 | 715 | 16.9 | 149 | 12.4 | 610 | 23.2 | 7454 | 24.3 | 735 | 22.9 | 120 | 21.2 | 243 | 14.6 | 15274 | 17.5 |
| 1975–1979 | 12604 | 29.4 | 1434 | 33.9 | 464 | 38.6 | 680 | 25.9 | 5831 | 19.0 | 513 | 16.0 | 60 | 10.6 | 423 | 25.4 | 22009 | 25.3 |
| 1980 | 21088 | 49.1 | 1639 | 38.7 | 502 | 41.8 | 298 | 11.4 | 2035 | 6.6 | 144 | 4.5 | 19 | 3.4 | 702 | 42.1 | 26427 | 30.3 |
| Missing | 3 | 0.0 | 0 | 0.0 | 0 | 0.0 | 0 | 0.0 | 6 | 0.0 | 1 | 0.0 | 0 | 0.0 | 8 | 0.5 | 18 | 0.0 |
| **Residence** | | | | | | | | | | | | | | | | | | |
| Urban | 21571 | 50.2 | 2486 | 58.7 | 700 | 58.3 | 1241 | 47.3 | 15867 | 51.6 | 1732 | 54.0 | 359 | 63.4 | 808 | 48.5 | 44764 | 51.4 |
| Rural | 21098 | 49.1 | 1729 | 40.8 | 495 | 41.2 | 1359 | 51.8 | 14621 | 47.6 | 1427 | 44.5 | 199 | 35.2 | 820 | 49.2 | 41748 | 47.9 |
| Missing | 262 | 0.6 | 20 | 0.5 | 6 | 0.5 | 24 | 0.9 | 233 | 0.8 | 47 | 1.5 | 8 | 1.4 | 39 | 2.3 | 639 | 0.7 |
| **Personal Income** | | | | | | | | | | | | | | | | | | |
| 7000 | 23166 | 54.0 | 2053 | 48.5 | 541 | 45.0 | 818 | 31.2 | 6966 | 22.7 | 1019 | 31.8 | 140 | 24.7 | 933 | 56.0 | 35636 | 40.9 |
| 7001–20000 | 16543 | 38.5 | 1810 | 42.7 | 587 | 48.9 | 1446 | 55.1 | 17513 | 57.0 | 1633 | 50.9 | 294 | 51.9 | 542 | 32.5 | 40368 | 46.3 |
| 20001 | 1927 | 4.5 | 275 | 6.5 | 50 | 4.2 | 321 | 12.2 | 5752 | 18.7 | 450 | 14.0 | 110 | 19.4 | 71 | 4.3 | 8956 | 10.3 |
| Missing | 1295 | 3.0 | 97 | 2.3 | 23 | 1.9 | 39 | 1.5 | 490 | 1.6 | 104 | 3.2 | 22 | 3.9 | 121 | 7.3 | 2191 | 2.5 |
| **Smoking status** | | | | | | | | | | | | | | | | | | |
| Never | 33659 | 78.4 | 2863 | 67.6 | 809 | 67.4 | 1648 | 62.8 | 19035 | 62.0 | 2135 | 66.6 | 291 | 51.4 | 712 | 42.7 | 61152 | 70.2 |
| Former | 4950 | 11.5 | 732 | 17.3 | 224 | 18.7 | 570 | 21.7 | 7181 | 23.4 | 583 | 18.2 | 169 | 29.9 | 293 | 17.6 | 14702 | 16.9 |
| current | 3550 | 8.3 | 567 | 13.4 | 148 | 12.3 | 293 | 11.2 | 3533 | 11.5 | 359 | 11.2 | 90 | 15.9 | 192 | 11.5 | 8732 | 10.0 |
| Missing | 772 | 1.8 | 73 | 1.7 | 20 | 1.7 | 113 | 4.3 | 972 | 3.2 | 129 | 4.0 | 16 | 2.8 | 470 | 28.2 | 2565 | 2.9 |
| **Alcohol consumption** | | | | | | | | | | | | | | | | | | |
| Occasional social drinker | 24676 | 57.5 | 2648 | 62.5 | 831 | 69.2 | 1607 | 61.2 | 18463 | 60.1 | 1877 | 58.5 | 345 | 61.0 | 924 | 55.4 | 51371 | 58.9 |
| Never drinker | 13181 | 30.7 | 902 | 21.3 | 182 | 15.2 | 559 | 21.3 | 6818 | 22.2 | 715 | 22.3 | 84 | 14.8 | 271 | 16.3 | 22712 | 26.1 |
| Current regular drinker | 1243 | 2.9 | 251 | 5.9 | 65 | 5.4 | 144 | 5.5 | 2187 | 7.1 | 173 | 5.4 | 54 | 9.5 | 54 | 3.2 | 4171 | 4.8 |
| Now stopped | 3416 | 8.0 | 404 | 9.5 | 114 | 9.5 | 267 | 10.2 | 2848 | 9.3 | 387 | 12.1 | 78 | 13.8 | 210 | 12.6 | 7724 | 8.9 |
| Missing | 415 | 1.0 | 30 | 0.7 | 9 | 0.7 | 47 | 1.8 | 405 | 1.3 | 54 | 1.7 | 5 | 0.9 | 208 | 12.5 | 1173 | 1.3 |
| **Pre-existing psychological conditions** | | | | | | | | | | | | | | | | | | |
| Not Having | 41365 | 96.4 | 4083 | 96.4 | 1157 | 96.3 | 2546 | 97.0 | 29875 | 97.2 | 3013 | 94.0 | 536 | 94.7 | 1585 | 95.1 | 84160 | 96.6 |
| Having | 1566 | 3.6 | 152 | 3.6 | 44 | 3.7 | 78 | 3.0 | 846 | 2.8 | 193 | 6.0 | 30 | 5.3 | 82 | 4.9 | 2991 | 3.4 |

**Table 4** Survival status (%) by sex, union status, family support and potential confounders for the Thai Cohort Study, 2005–2016

| | Males | | Females | | Total | |
|---|---|---|---|---|---|---|
| | **Survival status on December 201** | | | | | |
| | **Alive (%)** | **Dead (%)** | **Alive (%)** | **Dead (%)** | **Alive (%)** | **Dead (%)** |
| No of death | 38 507 | 978 | 47 235 | 423 | 85 750 | 1401 |
| Birth year | | | | | | |
| 1959 | 93.0 | 7.0 | 97.9 | 2.1 | 94.6 | 5.4 |
| 1960–1969 | 97.3 | 2.7 | 98.6 | 1.4 | 97.9 | 2.1 |
| 1970–1974 | 98.1 | 1.9 | 99.1 | 0.9 | 98.6 | 1.4 |
| 1975–1979 | 98.3 | 1.7 | 99.3 | 0.7 | 98.9 | 1.1 |
| 1980 | 98.2 | 1.8 | 99.4 | 0.6 | 99.0 | 1.0 |
| Missing | 100.0 | 0.0 | 100.0 | 0.0 | 100.0 | 0.0 |
| Residence | | | | | | |
| Urban | 97.7 | 2.3 | 99.1 | 0.9 | 98.5 | 1.5 |
| Rural | 97.3 | 2.7 | 99.1 | 0.9 | 98.3 | 1.7 |
| Missing | 95.7 | 4.3 | 99.4 | 0.6 | 97.7 | 2.3 |
| Personal income | | | | | | |
| 7000 | 97.6 | 2.4 | 99.2 | 0.8 | 98.6 | 1.4 |
| 7001–20000 | 97.7 | 2.3 | 99.1 | 0.9 | 98.4 | 1.6 |
| 20001 | 97.2 | 2.8 | 98.8 | 1.2 | 97.9 | 2.1 |
| Missing | 95.1 | 4.9 | 98.9 | 1.1 | 97.2 | 2.8 |
| Smoking status | | | | | | |
| Never | 98.2 | 1.8 | 99.1 | 0.9 | 98.9 | 1.1 |
| Former | 97.3 | 2.7 | 99.0 | 1.0 | 97.6 | 2.4 |
| Current | 96.4 | 3.6 | 97.5 | 2.5 | 96.5 | 3.5 |
| Missing | 96.4 | 3.6 | 99.1 | 0.9 | 98.1 | 1.9 |
| Alcohol consumption | | | | | | |
| Occasional social drinker | 97.9 | 2.1 | 99.2 | 0.8 | 98.5 | 1.5 |
| Never drinker | 97.8 | 2.2 | 99.1 | 0.9 | 98.8 | 1.2 |
| Current regular drinker | 96.6 | 3.4 | 98.1 | 1.9 | 96.7 | 3.3 |
| Now stopped | 95.8 | 4.2 | 98.8 | 1.2 | 97.1 | 2.9 |
| Missing | 96.2 | 3.8 | 99.4 | 0.6 | 98.2 | 1.8 |
| Pre-existing psychological conditions | | | | | | |
| Not having psychological diseases | 97.6 | 2.4 | 99.1 | 0.9 | 98.4 | 1.6 |
| Having psychological diseases | 96.0 | 4.0 | 98.7 | 1.3 | 97.6 | 2.4 |
| Pre-existing physical conditions | | | | | | |
| Not having diabetes | 97.6 | 2.4 | 99.1 | 0.9 | 98.5 | 1.5 |
| Having diabetes | 90.5 | 9.5 | 94.6 | 5.4 | 91.7 | 8.3 |
| Not having cancer | 97.5 | 2.5 | 99.1 | 0.9 | 98.4 | 1.6 |
| Having cancer | 94.0 | 6.0 | 95.9 | 4.1 | 95.2 | 4.8 |
| Not having CVD | 97.6 | 2.4 | 99.1 | 0.9 | 98.4 | 1.6 |
| Having CVD | 92.7 | 7.3 | 96.6 | 3.4 | 94.4 | 5.6 |
| Not having kidney/liver diseases | 97.6 | 2.4 | 99.2 | 0.8 | 98.5 | 1.5 |
| Having kidney/liver diseases | 96.3 | 3.7 | 98.0 | 2.0 | 97.0 | 3.0 |
| BMI | | | | | | |
| Underweight (<18.5) | 97.3 | 2.7 | 99.2 | 0.8 | 98.9 | 1.1 |

Continued

**Table 4** Continued

| | Males | | Females | | Total | |
|---|---|---|---|---|---|---|
| | Survival status on December 201 | | | | | |
| | Alive (%) | Dead (%) | Alive (%) | Dead (%) | Alive (%) | Dead (%) |
| Normal (18.5 to <23) | 97.7 | 2.3 | 99.2 | 0.8 | 98.6 | 1.4 |
| Overweight at risk (23 to <25) | 97.8 | 2.2 | 98.9 | 1.1 | 98.2 | 1.8 |
| Obese I (25 to <30) | 97.3 | 2.7 | 98.8 | 1.2 | 97.8 | 2.2 |
| Obese II (≥30) | 96.5 | 3.5 | 97.9 | 2.1 | 97.2 | 2.8 |
| Missing | 94.9 | 5.1 | 98.5 | 1.5 | 96.5 | 3.5 |
| Having good family support | | | | | | |
| Yes | 97.6 | 2.4 | 99.2 | 0.8 | 98.5 | 1.5 |
| No | 96.5 | 3.5 | 98.6 | 1.4 | 97.5 | 2.5 |
| Missing | 96.9 | 3.1 | 99.6 | 0.4 | 98.2 | 1.8 |
| Union status | | | | | | |
| Single | 98.0 | 2.0 | 99.2 | 0.8 | 98.7 | 1.3 |
| Cohabited and living with a partner | 97.9 | 2.1 | 98.9 | 1.1 | 98.5 | 1.5 |
| Cohabited but not living with a partner | 98.5 | 1.5 | 99.6 | 0.4 | 99.1 | 0.9 |
| Married but not living with a partner | 96.7 | 3.3 | 98.7 | 1.3 | 97.7 | 2.3 |
| Married and living with a partner | 97.3 | 2.7 | 99.1 | 0.9 | 98.1 | 1.9 |
| Separated/divorced/widowed | 94.3 | 5.7 | 98.8 | 1.2 | 97.3 | 2.7 |
| Separated/divorced/widowed but living with a partner | 95.7 | 4.3 | 98.2 | 1.8 | 96.6 | 3.4 |
| Missing | 97.3 | 2.7 | 98.7 | 1.3 | 98.0 | 2.0 |

BMI, body mass index; CVD, cardiovascular disease.

results of this subgroup analysis were in line with the main results. The cohabited group had a higher risk of dying among the younger age group (HR=1.54, p<0.05), when compared with the reference group. On the contrary, the higher risk of dying for persons who were separated, divorce or widowed was appeared in the older age group (HR=1.45, p<0.005). Results of this analysis were generally in line with the results among whole sample.

## DISCUSSION

The analyses add to Eastern population evidence on links between union status and mortality risk, as well as considering the underexplored factor of cohabitation. Single or cohabiting women had higher risks of mortality than women who were married and living together with a partner throughout follow-up. However, such increasing risk for single or cohabiting men was moderate. Separated/divorce/widowed men also had higher risks of mortality than men who were married and living together with a partner. Family support acts as a protective factor against mortality but such protective effect cannot fully be attributed to union status. These results confirm most of the hypotheses we proposed, except that cohabiting people do not appear to experience any protective effect over those who were single. In regard to the protective effect of marriage on mortality, this analysis agrees with much of the literature, and with similar effect sizes. For example in a meta-analysis, relative risks for mortality were calculated for widowed (1.11), divorced/separated (1.16) and never married (1.11) individuals, respectively.[1] Other studies however have found that cohabiting has a similar protective effect to marriage, unlike this Thai study. In fact in several studies there was no independent effect found for marriage on mortality, the protective function appeared to result from living together in any union status.[13 14] More complex analyses which take into account a range of factors around social interaction appear to predict mortality more accurately than models that use union status alone.[39]

A possible explanation for this divergence in results, may lie with the particular cultural setting of Thailand, and the lack of studies in similar settings. Defining and measuring marital status and cohabitation rates has historically been quite difficult in Thailand.[32] Buddhist society has not traditionally had a definite view on when couples are married and this has led to legal issues surrounding inheritance, property and alimony. Generally a man's obligation to treat a woman as his wife was defined through the payment of a dowry by the woman's family, without legal registration.[45]

A Family Registration Act in 1935 attempted to formalise marriage registration.[45] However, in modern Thailand multiple combinations of religious ceremony and state registration are still generally recognised[33]; and the Thai

**Table 5** Gender difference, Union status (2005)†, family support (2005) and mortality (2005–2016) in the Thai Cohort Study, 2005–2016

| Sex | | No of cases included in the model (% of total sample by sex) | Union status | | | | | | Not having good family support |
| --- | --- | --- | --- | --- | --- | --- | --- | --- | --- |
| | | | Single | Cohabited and living with partner | Cohabit but not living with partner | Married but not living together | Separated, divorce or widowed | Separated, divorced, widowed, but living with partner | |
| Total | 1 | 84 874 (97.4) | 1.40 (1.21–1.61)*** | 1.50 (1.15–1.96)*** | 0.98 (0.53–1.78) | 1.37 (1.05–1.79)* | 1.73 (1.38–2.18)*** | 1.46 (0.92–2.34) | -- |
| | 2 | 84 444 (96.9) | 1.36 (1.17–1.57)*** | 1.46 (1.12–1.90)** | 0.95 (0.52–1.74) | 1.35 (1.03–1.76)* | 1.65 (1.31–2.08)*** | 1.39 (0.87–2.22) | 1.39 (1.19–1.62)*** |
| | 3 | 82 461 (94.6) | 1.28 (1.11–1.49)*** | 1.43 (1.09–1.87)* | 0.95 (0.52–1.73) | 1.27 (0.97–1.67) | 1.48 (1.16–1.88)*** | 1.16 (0.70–1.95) | 1.38 (1.18–1.61)*** |
| | 4 | 79 286 (91.0) | 1.27 (1.09–1.48)*** | 1.39 (1.06–1.83)* | 0.96 (0.52–1.76) | 1.24 (0.92–1.66) | 1.46 (1.13–1.87)*** | 1.22 (0.73–2.05) | 1.35 (1.15–1.59)*** |
| Males | 1 | 38 390 (97.2) | 1.36 (1.13–1.63)*** | 1.34 (0.96–1.87) | 1.05 (0.52–2.12) | 1.32 (0.97–1.80) | 2.18 (1.67–2.85)*** | 1.39 (0.82–2.37) | -- |
| | 2 | 38 169 (96.7) | 1.32 (1.09–1.58)*** | 1.30 (0.93–1.82) | 1.03 (0.51–2.09) | 1.29 (0.94–1.77) | 2.09 (1.59–2.74)*** | 1.33 (0.78–2.27) | 1.30 (1.08–1.57)** |
| | 3 | 37 325 (94.5) | 1.20 (0.99–1.45) | 1.23 (0.88–1.73) | 0.99 (0.49–2.01) | 1.19 (0.86–1.64) | 1.87 (1.41–2.48)*** | 1.04 (0.57–1.90) | 1.30 (1.08–1.57)** |
| | 4 | 35 907 (90.9) | 1.17 (0.95–1.42) | 1.17 (0.83–1.65) | 0.99 (0.49–2.02) | 1.14 (0.80–1.62) | 1.79 (1.33–2.40)*** | 1.11 (0.61–2.03) | 1.29 (1.06–1.57)** |
| Females | 1 | 46 484 (97.5) | 1.42 (1.11–1.81)** | 1.87 (1.19–2.94)** | 0.81 (0.26–2.58) | 1.51 (0.90–2.54) | 1.12 (0.72–1.73) | 1.78 (0.66–4.82) | -- |
| | 2 | 46 275 (97.1) | 1.39 (1.09–1.78)** | 1.80 (1.15–2.82)* | 0.79 (0.25–2.49) | 1.49 (0.88–2.50) | 1.07 (0.69–1.66) | 1.65 (0.61–4.47) | 1.60 (1.21–2.10)*** |
| | 3 | 45 136 (94.7) | 1.40 (1.10–1.80)** | 1.87 (1.19–2.93)** | 0.83 (0.26–2.63) | 1.52 (0.90–2.56) | 0.98 (0.62–1.55) | 1.70 (0.63–4.61) | 1.56 (1.18–2.06)*** |
| | 4 | 43 379 (91.0) | 1.40 (1.08–1.81)** | 1.87 (1.19–2.96)** | 0.88 (0.28–2.78) | 1.49 (0.87–2.56) | 0.99 (0.62–1.59) | 1.75 (0.64–4.75) | 1.52 (1.14–2.03)*** |

Model 1 included union status+age+sex+urban (family support not included). Model 2 included union status+family support+age+sex+urban. Model 3 included union status+family support+age+sex+urban+income. Model 4 included union status+family support+age+sex+urban+income+ health behaviour +doctor diagnosed depression or anxiety+pre-existing physical conditions.
*p<0.05, **p<0.01, ***p<0.001.
†Reference group: married people and living together.

census itself may classify as married individuals who have only performed ceremonial bonding.[31] This is pertinent to our study with the possibility that some of those who self-reported being married may in fact according to western definitions be living in a cohabiting relationship. In Thailand cohabiting remains legally distinguishable from marriage and does not appear to confer mortality benefit but this needs more focused research to explain our results.

A potential explanation of high mortality risk for single and cohabited women living with a partner may be a consequence of social stigmatisation towards single or cohabitation status.[46 47] Social stigma towards single and cohabited persons and consequent stress of not being married may adversely affect persons' well-being and health status, in a society with a high prevalence of marriage.[46] In a society where most individuals are married, unmarried individuals have been seen as deviant from social role expectations.[46] For women, a 'marriage package' even comes laden with more traditional expectations of and for women.[48 49] Behaving against the norm may cause single and cohabitated people to evaluate their lives as more incomplete than married persons.[46] As well as this potential negative effect of not marrying, the converse is also possible. Marrying can have positive effects in terms of the instrumental and emotional support provided through marriage, and the improved sense of belonging.[50]

Results of our analysis showed that the mortality risk differentials for single males reduced and were no longer significant once income was included in the model. In contrast, the same adjustment had a limited impact for women. There is some evidence that men's low economic position would deter union formation, and that effect was more influential in countries where gender roles are more traditional.[51] Low-income men who are unable to fulfil the role of breadwinner are not only less attractive for marriage,[51] but their low socioeconomic status also leads to a higher risk of mortality.[52] Further, in a traditional society (like Thailand), low-income men who failed to meet the social norm of 'male breadwinner' may suffer from higher psychological distress.[53]

The gender difference in effect of marriage dissolution is also noteworthy. Separated/divorced/widowed men had much higher mortality risk than women. This result has also been found in previous studies, with the differential reducing in older age groups; that is middle-aged divorced or widowed men have the highest relative mortality risk.[6 54] One proposed mechanism for this differential is a greater reduction in social ties and social support experienced by men when marriages end.[3] With women being more responsible for maintaining friendship and family ties they are more capable to maintain these relationships after the marriage ends.[3]

The analysis included living arrangement combined with union status. It showed those married but living separately with a partner generally have a higher risk than those were married and living together, though the

effect is not statistically significant in the final model. It is possible that those who died during 2005–2016 were ill or at greater risks of mortality at baseline. There were 9353 participants (10.7% of cohort) reporting having physician's diagnosis of depression/anxiety, cancer, cardiovascular disease, diabetes, liver disease or kidney diseases at baseline. We performed a sensitivity analysis by excluding these participants from the survival analysis (see online supplemental appendix table 2) and found HRs for union status and family support were generally similar to the final model (model 4), which has adjusted for participants' baseline disease profile.

There were 79295 (91% among 87 151) cohort members aged from 20 to 44 years. A sensitivity analysis was conducted to compare the final model (model 4) among those aged 20–44 years with the model weighted by nationally representative Thai population. The national data were derived from the Health and Welfare Survey 2005[55] conducted and reported by the National Statistical Office (Thailand).[54] The HRs for union status and family support between the original model and model weighted by the Thai population were generally similar (online supplemental appendix table 3).

Our large nationwide cohort and extensive follow-up period produced reliable estimate of effect sizes while also measuring the longitudinal effects of changes in union status. Although the TCS participants are not randomly representative in terms of educational attainment, the majority Buddhist religion, geographical distribution, variance in socioeconomic well-being, as well as their community embedded residences give a reasonable picture of union status and mortality in Thailand.[40] The TCS is also future focused, including aspirational self-motivated Thais with increasing levels of education. Furthermore, unlike cross-sectional studies, longitudinal cohort analyses are designed to investigate causal relationships between exposures and outcomes within the same group of cohort population over time. Selection bias when establishing the cohort should not interfere with relative risk estimates between exposed and unexposed individuals for the sociobiological outcome of death.[55] HRs estimated from groups within a cohort remain valid and can be generalised more broadly.[56 57] In addition, the difficulty in defining marriage and cohabitation in Thailand may indicate there was some misclassification of individuals making the cohabitation—mortality relationship difficult to measure.

The study also has some limitations. First, for those who not having good family support, the mechanism for the effect of cohabitation on mortality may be different. Among those not having good family support, no significant mortality risk difference was found in those cohabited and living with a partner compared with the reference group (ie, who married living together with a partner). With limited support from family, cohabited individuals may face comparably less pressure from social norms but enjoy benefit (eg, emotional and financial support) from the relationship.[45] However, due to the

small number of deaths (n=) from this specified group, we did not explore further analysis. Also we also cannot be certain that the 'family support' variable does not also measure support from spouses or partners as well as other family members, though the questionnaire does use the Thai word implying wider family.

Second, we only have cause-of-death (COD) data for 583 deceased participants from 2005 to December 2010, accounting for 41.6% of all deaths. However, given the limited sample size (eg, only 78 CVD deaths, 118 neoplasms deaths and 204 injury deaths), it is not sufficient to assess the statistical association between union status and mortality by COD in the present analysis. Further analysis could be conducted in future research.

Third, we are unable to test the temporal alignments between the marriage status and the onset of the health condition that led to mortality. This means we cannot identify more precisely at what point in disease progression marital status exerts an influence.

Although this paper reports on data collected in Thailand, the results do have broader significance. Across Southeast Asia similar trends in changes in marriage and cohabitation are being observed. Female labour force participation and higher education participation along with changes in societal values are driving delayed marriage and increased cohabitation without marriage region wide.[58 59] This means that the associations we have measured here will have implications for other countries in the region experiencing similar transitions.

## CONCLUSIONS

Our 11-year cohort study revealed the protective effect of marriage on mortality in transitional Southeast Asia, an understudied Eastern setting. But such a protective effect of cohabitation, particularly for women, was limited in the Thai context. The gender difference found in this analysis suggests that union status effect on mortality may be mediated by sociocultural factors in a society, which deserves further investigation. The need to measure marriage and cohabitation in institutionally and culturally relevant terms is important. Formulation and implementation of public policies for moderating mortality should be gender nuanced and culturally and institutionally specific. The correct policy will respond to the current situation and encourage people to be aware of the potential effect of union status on health and well-being.

Thailand, as with most Southeast and East Asian countries, is undergoing a transition towards increased singlehood, delayed marriage, decreased family sizes and an ageing population. The findings of this paper in terms of the protective effect of marriage on health then have important implications for consideration of policies to promote healthy ageing. Healthy ageing policies cannot promote marriage, but can be sensitive to the increasing number of single and cohabiting individuals and their increased mortality risk.

**Author affiliations**

¹School of Demography, Australian National University, Canberra, Australian Capital Territory, Australia

²NHMRC Clinical Trials Centre, Faculty of Medicine and Health, The University of Sydney, Sydney, New South Wales, Australia

³Department of Global Health, Australian National University, Canberra, Australian Capital Territory, Australia

⁴Australian Research Council Centre of Excellence on Population Ageing Research, Business School, University of New South Wales, Sydney, New South Wales, Australia

⁵Sukhothai Thammathirat Open University, Bang Phut, Thailand

⁶National Centre for Epidemiology and Population Health, Australian National University, Canberra, Australian Capital Territory, Australia

**Acknowledgements** We thank the staff at Sukhothai Thammathirat Open University (STOU) who assisted with student contact and the STOU students who are participating in the cohort study. We also thank Dr Bandit Thinkamrop and his team from Khon Kaen University for guiding us successfully through the complex data processing.

**Contributors** JZ and CKL conceived the study design and led data analysis, MK and VY led data interpretation and manuscript writing. AS and S-AS conceived the overall study and made substantial contributions to data interpretation and critically reviewing the manuscript. All authors contributed to review and approval of the final draft. Mk accepts full responsibility for the work and/or the conduct of the study, had access to the data, and controlled the decision to publish.

**Funding** This study was supported by the International Collaborative Research Grants Scheme with joint grants from the Wellcome Trust UK (GR071587MA) and the Australian NHMRC (268055) and by a global health grant from the NHMRC (585426).

**Competing interests** None declared.

**Patient and public involvement** Patients and/or the public were involved in the design, or conduct, or reporting, or dissemination plans of this research. Refer to the Methods section for further details.

**Patient consent for publication** Not applicable.

**Ethics approval** Ethics approval was obtained from Sukhothai Thammathirat Open University Research and Development Institute (protocol 0522/10) and the Australian National University Human Research Ethics Committee (protocols 2004/344 and 2009/570). Participants gave informed consent to participate in the study before taking part.

**Provenance and peer review** Not commissioned; externally peer reviewed.

**Data availability statement** Data are available on reasonable request. Data can be shared by communication with corresponding author.

**ORCID iDs**

Matthew Kelly http://orcid.org/0000-0001-7963-2139

Adrian Sleigh http://orcid.org/0000-0001-8443-7864

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
