## [Reviewer comments · BMJ Open]

ARTICLE DETAILS

TITLE (PROVISIONAL)	How do cohabitation and marital status, affect mortality risk?: results from a cohort study in Thailand
AUTHORS	Zhao, Jiaying; Law, CK; Kelly, Matthew; Yiengprugsawan, Vasoontara; Seubsman, Sam-ang; Sleight, Adrian

VERSION 1 – REVIEW

REVIEWER	Roth, Adam Indiana University
REVIEW RETURNED	14-Apr-2022

GENERAL COMMENTS	This paper the role of union status on mortality in Thailand. It is well written and easy to follow. Please see below for my comments. 1. There could be a bit more motivation in exploring family support, especially the interaction between family support and union status (see for example Roth and Peng 2022). On the note of the family support x union status interaction, I did not see it in the model yet the authors report on it in the text by saying "Overall, no significant interaction was found between union status and family support ($p>0.05$), except for those who cohabited and were living with a partner. Among respondents who had poor family support, no mortality risk difference was found in those cohabited and living with a partner ($p=0.04$)" on page 24. Is that interpreted correctly? If there was a significant interaction for those who cohabited and were living with a partner, shouldn't there be a significant difference in mortality risk? Presenting predicted probabilities or a visual might help here. Roth, A. R., & Peng, S. (2022). Non-Spousal Support, Marital Status, and Mortality Risk. Journal of Aging and Health, 34(1), 41–50. https://doi.org/10.1177/08982643211025381 2. Why did the authors dichotomize the family support variable? What was the exact distribution of this variable for the entire sample?3. It seems that it would be useful to know if the family support question included support from one's spouse or only non-spousal support since this could matter a lot for married vs. non-married respondents. The authors may not have this information, but at minimum it should be noted as a limitation.4. Do the authors have data on cause of death? An all-cause mortality study is certainly valid, but breaking it down by cause of death could provide better insight into the specific pathways
---

	through which union status (and support) influences how men and women die. 5. The sample is very young. Nearly half of the sample was born in 1980 or later, which would make them 36 years or younger by 2016. This seems like it could cause some issues. First, the mortality rate is very low for younger people. Why not just focus on those who are considerably older? Also depending on the low end of the age range, it might be that younger people--who are themselves less likely to die during follow-up, might also be less likely to be married, divorced, widowed, etc. because they haven't reach that stage of their life yet. Finally, younger people tend to die from different causes of death than older people (e.g., accidents, overdoses, etc. vs strokes, Alzheimer's disease, heart failure). To my previous point without being able to breakdown the sample by cause of mortality, this might further complicate the findings if you are to include the younger cohorts in the analysis. 6. The authors mention on pg. 28 that social stigmatisation could be responsible for the finding that married people have lower mortality risk than non-married. But is this the only possibility? What about sense of belonging/commitment to their partner/family? That seems like another potential mechanism that is at least worth mentioning in the Discussion.
--	--

REVIEWER	Nakaya, N Tohoku University
REVIEW RETURNED	23-Apr-2022

GENERAL COMMENTS	Although this is a classical epidemiological research, it is an interesting study that attempts to analyze social factors (marital status) and health using large-scale Thai data. On the other hand, I believe that the scientific value will be further enhanced by considering the following issues. 1. The author's hypothetical mechanism is unclear from the statistical model of this study. The authors made four statistical models and interpreted the results, but it is unclear whether the items put into the model are "confounding" and "intermediating factors". Furthermore, what kind of mechanism do authors think that the statistically significant difference remains even after all variables adjustment? Perhaps the authors envision biological mechanisms (eg psychoneuroimmunologic responses and/or poor lifestyles). But isn't it a residual confounding? The authors need a clear hypothetical mechanism. Furthermore, since the cause of death is unknown, the reader cannot tell what kind of pathway the results of this study are through. The authors have all-cause mortality as an outcome, but what are their hypotheses about cancer, cardiovascular disease, suicide, road accidents, drowning, and so on? This study is a straightforward result, but it cannot be interpreted in the current manuscript. Please justify the above issues. 2. When considering in the results of this study, it is extremely difficult to intervene in marriage status. If the authors think that intervening in this is no longer within the scope of medical research. I don't think it makes sense to encourage single (cohabited) people to get married. That's because marriage status is their choice. How does this difficult-to-intervene study contribute to medicine? Will it increase income for poor single (cohabited) people? Or do you offer a family support? That is impossible. A
---

	single (cohabited) person may not be able to get married due to various health (physical, mental, social) problems. Justify the clinical significance of this study. 3. Since this study has mortality as an outcome, it is unclear whether marriage status influences the onset of the disease or the prognosis of life from the disease. Although it is related to 1. above, I feel that the interpretation is complicated by making death an outcome. Please justify this issue as well. 4. Will the marriage status not change during the 11-year follow-up? It may be the situation in Thailand, but if the change cannot be tracked, the results of this study may be underestimated. Consider its impact. 5. Is the result of this research a result that can be generalized in the world? Consider why.
--	---

VERSION 1 – AUTHOR RESPONSE

Reviewer 1:

1. There could be a bit more motivation in exploring family support, especially the interaction between family support and union status (see for example Roth and Peng 2022). On the note of the family support x union status interaction, I did not see it in the model yet the authors report on it in the text by saying "Overall, no significant interaction was found between union status and family support ($p > 0.05$), except for those who cohabited and were living with a partner. Among respondents who had poor family support, no mortality risk difference was found in those cohabited and living with a partner ($p = 0.04$)" on page 24. Is that interpreted correctly? If there was a significant interaction for those who cohabited and were living with a partner, shouldn't there be a significant difference in mortality risk? Presenting predicted probabilities or a visual might help here.

Roth, A. R., & Peng, S. (2022). Non-Spousal Support, Marital Status, and Mortality Risk. *Journal of Aging and Health*, 34(1), 41–50. <https://doi.org/10.1177/08982643211025381>

Response:

We reported the interaction between union status and family support status in Appendix Tables 1 and 2. We stratified analysis by whether participants had a good family support at baseline.

Page 23, Line 7:

We further discussed this implication and acknowledge as a limitation in Discussion.

Page 30, Last paragraph

2. Why did the authors dichotomize the family support variable? What was the exact distribution of this variable for the entire sample? (MK, JZ)

Response: We have reported the original distribution of family support by original questions and death. The proportion of death during 11 year follow up for those 'quite a bit of support', and 'a lot of support' was 1.65% and 1.45%, while that for those "very little", 'little', or 'Not relevant' was 2.67%, 2.24% and 3.75% respectively. The statistical power may be influenced due to the small number of deceased participants for those with little (128), very little (57) support and not relevant (23), and a

merger of the three groups enables a more robust statistical analysis in the present study. We added the rationale for dichotomize on Page 13 in the 'family support' section.

3. It seems that it would be useful to know if the family support question included support from one's spouse or only non-spousal support since this could matter a lot for married vs. non-married respondents. The authors may not have this information, but at minimum it should be noted as a limitation.

Response: Thank you for the comments. Unfortunately, we can't distinguish the family support from one's spouse or non-spousal support. We have added this as a limitation on Page 31.

4. Do the authors have data on cause of death? An all-cause mortality study is certainly valid, but breaking it down by cause of death could provide better insight into the specific pathways through which union status (and support) influences how men and women die. (JZ do the analysis first)

Response: Thanks for that. Unfortunately, we only have cause of death data to 2013. There were only 583 deaths with cause in our data set, accounting for 41.6% of all deaths, with limited time of follow up. We prepared a table to report the number of death by cause of death and union status, and added it in Appendix Table 3. However, due to limited sample size (e.g. only 78 cardiovascular death, e.g. only 78 cardiovascular deaths, 118 neoplasms deaths, and 204 injury deaths), it is insufficient to assess the statistical associations between union status and mortality by cause of death in the present analysis.

We have discussed the cause of death issue in the limitations section of the Discussion page 31

5. The sample is very young. Nearly half of the sample was born in 1980 or later, which would make them 36 years or younger by 2016. This seems like it could cause some issues. First, the mortality rate is very low for younger people. Why not just focus on those who are considerably older? Also depending on the low end of the age range, it might be that younger people--who are themselves less likely to die during follow-up, might also be less likely to be married, divorced, widowed, etc. because they haven't reach that stage of their life yet. Finally, younger people tend to die from different causes of death than older people (e.g., accidents, overdoses, etc. vs strokes, Alzheimer's disease, heart failure). To my previous point without being able to breakdown the sample by cause of mortality, this might further complicate the findings if you are to include the younger cohorts in the analysis .

Response: We agreed with reviewers' comments. We conducted a sensitivity analysis for cause of mortality by union status with 583 deaths. As mentioned in Question 4, we do not think that the death numbers by COD are sufficient to further stratification by age group and union status. We acknowledge it as a limitation on Page 31.

We acknowledged that older people are more likely to be married, divorced, widowed. We further conducted a sensitivity analysis by stratifying age group (those who were born before 1975 versus those who were born in 1975 or latter). It is found that the cohabited group had a higher risk of dying among the younger age group while those who were separated, divorce or widowed had a higher mortality risk for the older age group. Results of this analysis were generally in line with the main results.

Page 27:

6. The authors mention on pg. 28 that social stigmatisation could be responsible for the finding that married people have lower mortality risk than non-married. But is this the only possibility? What

about sense of belonging/commitment to their partner/family? That seems like another potential mechanism that is at least worth mentioning in the Discussion. (MK)

Response: We agree that it is useful to consider the positive influence of marriage as well as the potential negative influence of not marrying. We have added some extra text regarding the sense of belonging/commitment to their partner/family on page 28.

Reviewer 2:

7. The author's hypothetical mechanism is unclear from the statistical model of this study. The authors made four statistical models and interpreted the results, but it is unclear whether the items put into the model are "confounding" and "intermediating factors". Furthermore, what kind of mechanism do authors think that the statistically significant difference remains even after all variables adjustment? Perhaps the authors envision biological mechanisms (eg psychoneuroimmunologic responses and/or poor lifestyles). But isn't it a residual confounding? The authors need a clear hypothetical mechanism.

Response: we have added extra text on page 17 to explain our inclusion of variables in these models: The analysis process aimed to adjust confounding variables. From Model 1 to Model 3, we gradually add confounding variables at individual, family and socio-economic level. In Model 4, we included health behaviour and pre-existing psychological and physical condition and BMI. These health behaviour and status variables may serve as "intermediating factors" for the effect of union status on mortality if the HR for union status changed significantly when these variables were added in the Model. If HR for union status changed little, these health behaviour and status variables may act as confounding variables for the effect of union status on mortality [44].

8. Furthermore, since the cause of death is unknown, the reader cannot tell what kind of pathway the results of this study are through. The authors have all-cause mortality as an outcome, but what are their hypotheses about cancer, cardiovascular disease, suicide, road accidents, drowning, and so on? This study is a straightforward result, but it cannot be interpreted in the current manuscript. Please justify the above issues.

Response: We agree that different causes of death have their own pathways. Unfortunately, we only have cause of death data to 2013. There were only 583 deaths, accounting for 41.6% of all deaths, with limited time of follow up. We prepare a table to report the number of death by cause of death and union status, and added it in Appendix Table 3. However, due to limited sample size (e.g. only 78 cardiovascular deaths, 118 neoplasms deaths, and 204 injury deaths), it is insufficient to assess the statistical associations between union status and mortality by cause of death in the present analysis. The reporting of only all-cause mortality of course limits our ability to interpret the pathways by which marriage can connect with or protect from mortality, but the associations with all-cause mortality are still important in ascertaining overall risk.

9. When considering in the results of this study, it is extremely difficult to intervene in marriage status. If the authors think that intervening in this is no longer within the scope of medical research. I don't think it makes sense to encourage single (cohabited) people to get married. That's because marriage status is their choice. How does this difficult-to-intervene study contribute to medicine? Will it increase income for poor single (cohabited) people? Or do you offer a family support? That is impossible. A single (cohabited) person may not be able to get married due to various health (physical, mental, social) problems. Justify the clinical significance of this study. (MK, could we add some discussion for policy implication as reviewers' suggestion, and I will do the analysis for cohabited only in related Review 1's comment.)

Response: We agreed the reviewer's comment. It does not make sense to encourage single (cohabited) people to get married. However, union status is a marker to identify high risk individuals. We have added some text to the end of the Conclusion which considers the policy implications. Basically, in a context where governments are making policies to promote healthy ageing in ageing

societies it is important to consider the life course risk factors which affect risk of death in older age. This does not mean promoting marriage itself, but rather considering the additional health needs of those who do not marry.

10. Since this study has mortality as an outcome, it is unclear whether marriage status influences the onset of the disease or the prognosis of life from the disease. Although it is related to 1. above, I feel that the interpretation is complicated by making death an outcome. Please justify this issue as well.

Response: We agreed with your comments. Mortality is a final end point as health outcome. We do not collect the information of onset and progression of diseases. But it is a reasonable assumption to hypothesize that union status would affect the risk of these two events with a similar social and pathological mechanism. We now acknowledge this as a limitation of the study. We do not know whether marriage status preceded or influenced the onset of the disease. But, there is a statistically significant association between marriage status and the mortality outcome which is the main finding of our study.

11. Will the marriage status not change during the 11-year follow-up? It may be the situation in Thailand, but if the change cannot be tracked, the results of this study may be underestimated. Consider its impact. (MK see previous version, table 4)

Response: We agreed with the reviewers' comments. We have a follow up wave in 2009, with an overall response rate of 71% to track the participants' union status. We have conducted sensitivity analysis to examine changes in union status from 2005 to 2009 and death (Appendix Table 4).

The paper aims to examine the relationship between baseline union status and mortality, focus on cohabitation and gender differentials in Thailand. Due to the small number of deaths for participants who changed from or to cohabitation, it is insufficient to assess the statistical associations between changes in union status and mortality in the present analysis. Therefore, we did not put this in the main results.

12. Is the result of this research a result that can be generalized in the world? Consider why.

Response: We agree that the results are specific to Thailand. However the trends in terms of increasing cohabitation and delayed marriage are occurring across much of Asia. We have added a paragraph to the end of the discussion explaining how the results are more broadly applicable.

VERSION 2 – REVIEW

REVIEWER	Roth, Adam Indiana University
REVIEW RETURNED	14-Aug-2022

GENERAL COMMENTS	Thank you for the revisions.
------------------------------

REVIEWER	Nakaya, N Tohoku University
REVIEW RETURNED	01-Aug-2022

GENERAL COMMENTS	I have no further comments about this paper.
--